# Polarization curling and flux closures in multiferroic tunnel junctions

Jonathan J.P. Peters[1], Geanina Apachitei[1], Richard Beanland[1], Marin Alexe[1] & Ana M. Sanchez[1]

Formation of domain walls in ferroelectrics is not energetically favourable in low-dimensional systems. Instead, vortex-type structures are formed that are driven by depolarization fields occurring in such systems. Consequently, polarization vortices have only been experimentally found in systems in which these fields are deliberately maximized, that is, in films between insulating layers. As such configurations are devoid of screening charges provided by metal electrodes, commonly used in electronic devices, it is wise to investigate if curling polarization structures are innate to ferroelectricity or induced by the absence of electrodes. Here we show that in unpoled $Co/PbTiO_3/(La,Sr)MnO_3$ ferroelectric tunnel junctions, the polarization in active $PbTiO_3$ layers 9 unit cells thick forms Kittel-like domains, while at 6 unit cells there is a complex flux-closure curling behaviour resembling an incommensurate phase. Reducing the thickness to 3 unit cells, there is an almost complete loss of switchable polarization associated with an internal gradient.

[1] Department of Physics, University of Warwick, Gibbet Hill Road, Coventry CV4 7AL, UK. Correspondence and requests for materials should be addressed to J.J.P. (email: j.j.p.peters@warwick.ac.uk) or to M.A. (email: m.alexe@warwick.ac.uk).

I t has been recently shown, at all scales, that ferroelectric polarization forms a structure more complex than simple textbook Kittel type domains[1–4]. At the ferroelectric–ferroelastic domain walls, polarization curling is enhanced by the flexoelectric response to strain gradients[5]; whilst a continuous polarization rotation at non-ferroelastic domain walls has only been rarely reported[6,7]. Arguably, all reports on atomic scale curling of polarization and vortex structures are on systems without electrodes. Whilst this favours vortex structures through enhancement of depolarizing fields, manipulation and real use of these vortices necessitates metal electrodes to apply electric fields. The screening effect of free carriers in the electrodes will apply a different set of constraints to those found in previously studied vortex structures. This raises the fundamental question of the stability of these vortex structures in metal–ferroelectric–metal, that is, capacitor devices. To address this question, capacitors with active ferroelectric film thicknesses of the order of a few unit cells (u.c.) have to be fabricated. Fortunately, the above structures are ferroelectric tunnel junctions that allow quantum tunnelling between the two metal electrodes, as in well-known magnetic tunnel junctions[8]. Moreover, the tunnel current is a tool for non-destructive reading the polarization state. As has been shown, switching of the ferroelectric polarization modulates the tunnelling current by more than five orders of magnitude[9–11] and in combination with magnetic electrodes a multiferroic tunnel junction (MFTJ) is formed. This allows an extra degree of freedom so that MFTJs can be used for four-state memory devices, controlled by both magnetic and electric fields[12–14], or even controlling spin filtering via ferroelectric polarization[15].

We expect both the electrodes and their interfaces with the ferroelectric to play a critical role in formation of the domain structure. An electrode's ability to screen the depolarization field produced by bound charges at the surface of the ferroelectric is dependent of its free carrier concentration. An asymmetry in the electrodes may therefore create a more complex domain pattern[16]. Whilst the ingress of a depolarizing field into the ferroelectric film due to screening effects, which can easily reach values of the order of $1\,GV\,m^{-1}$, is opposed by the material irrespective of the type of domain, it is to be expected that the interplay between the elastic and electric field will favour polarization curling.

Here we use aberration-corrected scanning transmission electron microscopy (STEM) to study ferroelectric domains and domain walls in unpoled ferroelectric tunnel junctions with ultrathin active PbTiO$_3$ (PTO) ferroelectric layers (3, 6 and 9 u.c.)

sandwiched between Co and La$_{0.7}$Sr$_{0.3}$MnO$_3$ (LSMO) electrodes. STEM was used to analyse the domain structure of these devices in detail, imaging along the [110] axis of the PTO. Annular dark field (ADF) and annular bright field (ABF) images were collected simultaneously, enabling direct visualization of both heavy and light elements with atomic resolution. Thus, the ion positions and relative displacements can be measured, giving the dipole distribution. We obtain local dipole maps by considering the B-site ion displacement with respect to the centre of the O$^{2-}$ octahedra ($\Delta_B$, see the 'Methods' section). As the film thickness is reduced the ferroelectric domains undergo a transition between 3 regimes: from Kittel-like domains, as seen in thick films[17], through complex curling resembling an incommensurate phase[18] to monodomain material in the very thinnest layers. We find that, in contrast to thicker films where the domain walls are thin and relatively smooth[19], the domain walls in ultrathin ferroelectrics develop into topological entities with a flourishing domain structure involving polarization curling, spanning from Bloch(Néel)-Ising structures to flux closure and vortices.

## Results

**Multiferroic tunnel junction devices.** Epitaxial ferroelectric PTO films were deposited by reflection high-energy electron diffraction (RHEED)-assisted pulsed laser deposition (PLD) on LSMO buffered (100)-oriented SrTiO$_3$ (STO) single crystal substrates. Cobalt electrodes were then sputtered *ex situ* (details of the growth are provided in the 'Methods' section). Figure 1a shows a low-magnification ADF–STEM image of the MFTJ with a 6-u.c.-thick PTO layer. The corresponding tunnelling magneto-resistance (TMR) and tunnelling electroresistance (TER) are shown in Fig. 1b. For all thicknesses, except 3 u.c., application of an external electric field switches polarization and induces a significant TER effect, as previous studies have shown[10,20]. All of the investigated Co/PTO/LSMO structures show TMR effects.

**9-u.c.-thick PTO tunnel junction.** Figure 2a shows an ADF image from the 9 u.c. PTO sample, including the interfaces with the Co and LSMO electrodes. The ADF image is sensitive to the chemical composition of the sample as the atom column intensity scales as approximately $Z^{1.7}$ where $Z$ is the average atomic number of the atom column (although O$^{2-}$ are not visible)[21]. The intensity profile in Fig. 2b demonstrates the atomically sharp LSMO–PTO interface with Ti–O/La–Sr termination between the layers. Figure 2c shows the corresponding ABF image where

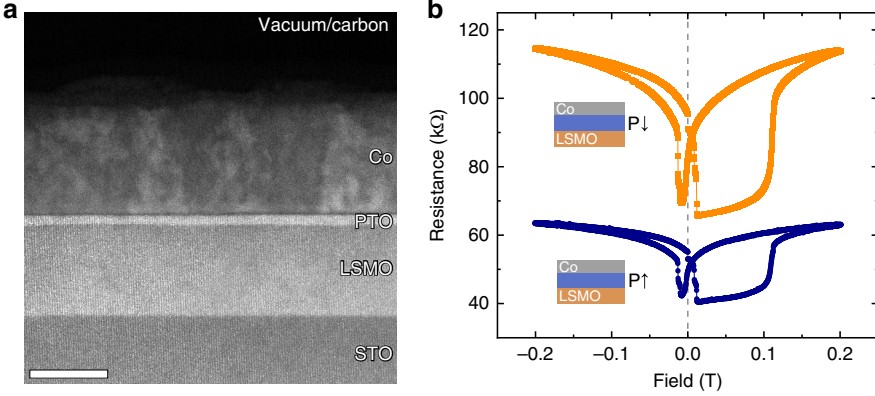

**Figure 1 | Device structure and electrical characterization. (a)** ADF–STEM image of the 6 unit cells thick MFTJ. The contrast shows the different layers. Scale bar, 20 nm. (**b**) TMR loops for both polarization towards Co (blue circles) and towards LSMO (orange squares). Resistance was measured at a 0.2 V bias after cooling to 10 K in a −0.8 T magnetic field.

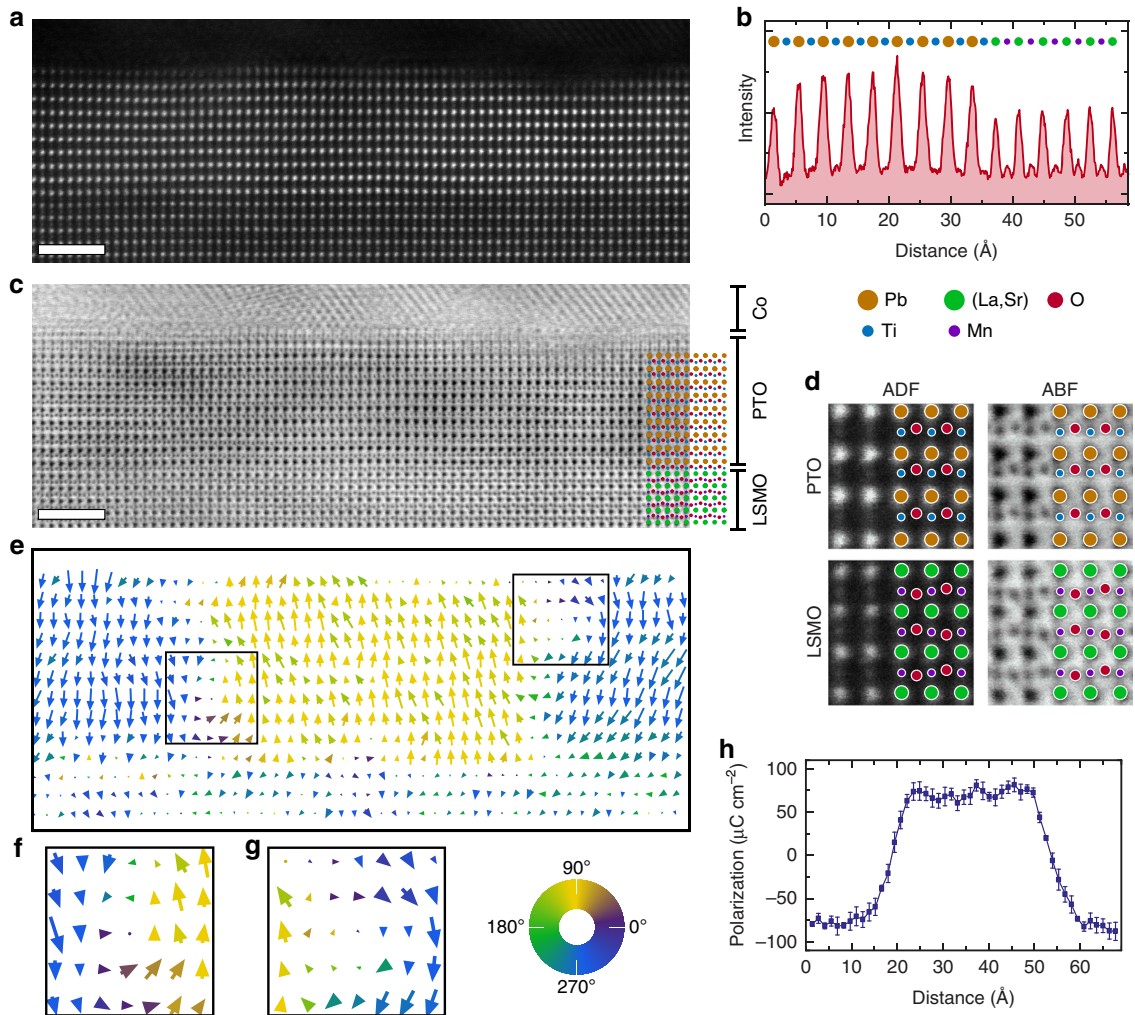

**Figure 2 | Nine unit cells thick PTO tunnel junction.** (**a**) Atomic resolution ADF image. (**b**) Intensity plot from **a** showing atom column composition. (**c**) ABF signal collected simultaneously to **a**. All scale bars, 2 nm. (**d**) Magnified regions from **a,c** showing the structure in LSMO and PTO and the difference in contrast. (**e**) Quiver plot showing dipoles measured from **c**. (**f,g**) Enlarged versions of the regions highlighted on the left and right domain walls in **e** respectively. Each area is centred on a vortex. (**h**) Average out of plane polarization across the domains. Error bars are the s.e.m.

the $O^{2-}$ columns are now visible and the B cation contrast has improved (Fig. 2d). From this image the column positions is measurable to a few picometres[22,23].

The two-dimensional quiver plot of the local dipole, shown in Fig. 2e, indicates that PTO is in a ferroelectric state. In the centre of the image, the polarization points upward towards the Co whereas on the left and right polarization points downwards, that is, a classic 180° domain structure. This would be expected to show Ising type domain walls with a wall width of the order of 2 u.c. Nevertheless, a relatively wide wall with a roughness of ~4–5 u.c. is revealed. Additionally, the polarization map provides evidence of two small vortices, highlighted in Fig. 2f,g. In these areas, the dipole direction changes continuously anti-clockwise then clockwise, forming a paired vortex and anti-vortex state similar to the PTO–STO superlattice case recently reported by Yadav *et al.*[7] The vortices are located at the domain walls, forming a more complex structure separating the two opposite ferroelectric domains. The clockwise vortex is observed close to the Co–PTO interface whilst the anti-clockwise vortex is near the LSMO–PTO interface. We note that an atom column that contains varying displacements perpendicular to the beam direction would appear elongated (that is, corresponding to the projection of the mixed positions). Such an effect is absent,

indicating that the measured dipoles are constant through the thickness of the specimen. This would not be the case with Néel or Bloch-type walls extending within the thickness of the TEM specimen. From Fig. 2 it can therefore be inferred that the depolarization field, always present in ferroelectric films, has an effect on the polarization distribution when the PTO film thickness is below 10 unit cells, though this may extend to even thicker films, as predicted by ab initio calculations[24]. It is important to note the asymmetry of the electrodes and how this affects polarization in the PTO ultrathin film. Vertically averaging the out of plane polarization gives Fig. 2h with the amplitude smoothly changing between positive and negative at the domain walls. The polarization pointing down towards the LSMO electrode has an average value of $80 \pm 1 \, \mu C \, cm^{-2}$, close to the value of $\sim 84 \, \mu C \, cm^{-2}$ obtained using bulk displacements[25]. There exists a decrease in the polarization magnitude pointing towards the Co ($70 \pm 2 \, \mu C \, cm^{-2}$), a result of the asymmetric screening of the depolarization field by the different electrodes. It can be seen in Supplementary Fig. 1 that the polarization is mostly constant across the film but extends into the LSMO layer, inducing a displacement within the first 2–3 u.c. of the interface. Cobalt, as a good metal, is able to screen the positive and negative charge accumulation, whereas LSMO is a half-metal with lower

carrier concentration than Co. (ref. 26) Figure 2h also shows the rougher/thicker (4–5 u.c.) domain walls from the observed in-plain (90°) component of the polarization vortices. This will have an effect on the domain wall motion and the piezoelectric coefficient.

**6-u.c.-thick PTO tunnel junction.** Comparable analyses were carried out in heterostructures contain 6 u.c. of PTO between similar LSMO and Co electrodes. ADF and ABF imaging, Fig. 3a,b respectively, demonstrate an excellent crystal quality with defect free atomically sharp interfaces. At first glance, both 9 and 6 u.c. look the same, but the quiver plot of the dipole in Fig. 3(c) indicates a drastic change in the domain pattern, although the PTO is still in a polarized state. The relatively well-defined domain walls (with associated small vortices) observed in 9 u.c. of PTO have been replaced with a structure showing a combination of Landau–Lifshitz flux-closure domains and vortices. However, over a large area (∼75 nm) there is a high degree

of disorder: areas where polarization switches from the top to bottom of the layer (Fig. 3e), complex regions of curling (Fig. 3f) and (110) type domain walls forming at 45° to the interface (Fig. 3g), typical of thicker films. The full data is shown in Supplementary Fig. 2. Some of these types of domain walls have been predicted by first-principles based methods for thicknesses corresponding to 6 u.c. For instance, closure at the bottom LSMO electrode may occur as a response to the asymmetry in the electrode screening[16]. The high degree of disorder with a certain remanent periodicity observed here suggests that the system actually enters into an incommensurate phase. It has been recently predicted that general ferroelastic systems in a certain range of dimensionality no longer undergo a simple para-ferro phase transition, instead entering, across the tri-critical Lifshitz point, an incommensurate phase[18]. The driving force for this is the flexoelectric effect. Indeed, our investigated PTO is both ferroelectric and ferroelastic and, under a certain thickness, additional parameters such as inhomogeneous electric fields might induce crossing of the Lifshitz point and drive the system

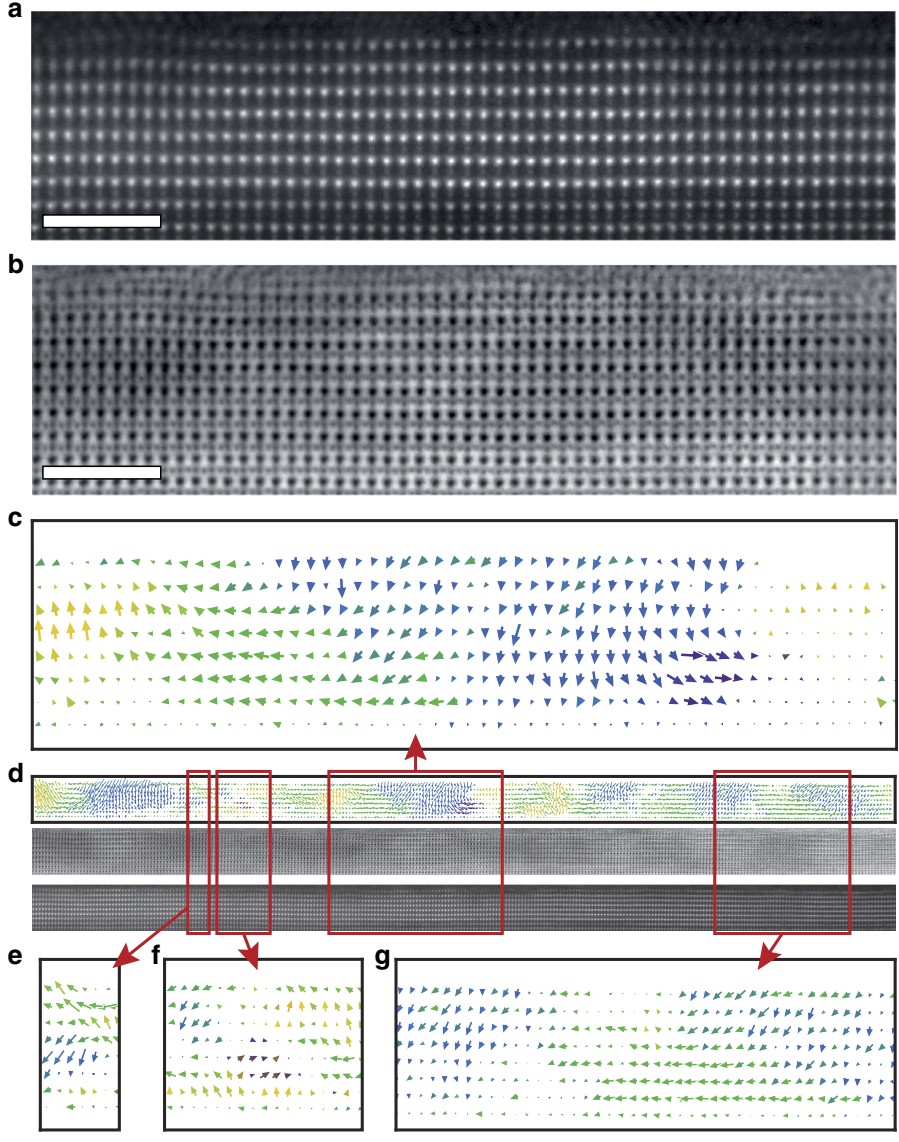

**Figure 3 | Six unit cells thick PTO tunnel junction.** (**a,b**) Atomic resolution ADF and ABF images respectively. All Scale bars, 2 nm. (**c**) Corresponding quiver plot of dipoles. (**d**) Large area analysis showing a dipole map, ABF and ADF images. An enlarged version is shown in Supplementary Fig. 2. (**e–g**) Regions of interest taken from the indicated regions of **d**.

into an incommensurate phase. Nevertheless, applying external electric field switches the polarization as expected for a proper ferroelectric film (the TER effect is ~81% as in Fig. 1b).

Interestingly, at the Co interface, the domain width can be up to ~8 nm, larger than in the 9 u.c. sample (6.4 ± 0.2 nm), though at the LSMO interface the domain reduces to ~4 nm. Other domains with parallel walls, 45° to the substrate, have constant widths across the thickness of the film and can be as small as ~3 nm. Thus the average of the measured widths is 5.3 ± 0.7 nm, in agreement with the expected value of 5.2 nm obtained from the Kittel scaling law with the 9 u.c. domain width[27].

**3-u.c.-thick PTO tunnel junction.** Further reducing the thickness of the ferroelectric film to 3 u.c. results in the domain structure largely disappearing in unpoled material (Fig. 4). Instead the polarization shows a significant gradient across the entire film thickness (Fig. 4c). Remanent curling is present, but polarization largely points from Co to LSMO. The average magnitude of the polarization at the Co interface is $76 \pm 5\,\mu C\,cm^{-2}$, the same as the equivalent measurement in 9 u.c. of $79 \pm 1\,\mu C\,cm^{-2}$, whilst the displacements at the LSMO interface are zero. This suggests that 3 u.c. is the lower limit of ferroelectricity in PTO under this positive in-plane misfit strain. The TER effect disappears which is an indication that the ferroelectric polarization cannot be switched anymore, likely due to its gradient within the ferroelectric layer. We note that this gradient in polarization must be associated with a gradient in electric field and therefore with free charge localized within the ferroelectric layer. As mentioned, the TER effect for the 3 u.c. layer is suppressed, but the TMR effect is still significant (as shown in Supplementary Fig. 3). This demonstrates that the tunnel junction device based on the 3-u.c. PTO layer is functional but the internal gradient causes polarization to become non-switchable.

## Discussion

In summary, we have shown that in PTO-based MFTJs the ferroelectric domain pattern for PTO at a thickness of 9 u.c. is generally classic antiparallel (180°) with Ising type domain walls decorated with coupled clockwise and anti-clockwise vortices. For 6 u.c., a peculiar domain pattern with curling flux-closure type structures and incommensurate phase was observed. This indicates a crossing of a tri-critical Lifshitz point due to coupling of the ferroelectric–ferroelastic properties of PTO with the flexoelectric effect. For only 3-u.c.-thick PTO films, domain structure is widely suppressed with polarization pointing out of plane and remanent domain structure.

The present results show that the polarization curling and formation of vortex and flux-closure structures is a generic effect that appears in ultrathin ferroelectric films, even with metal electrodes. The screening ability of such electrodes do not prevent these polarization curling structures from forming and the possibility to manipulate these structures with local electric fields is open.

## Methods

**Sample growth.** The oxide layers were grown by RHEED assisted PLD using a 248 nm wavelength KrF excimer laser. Atomically smooth vicinal surfaces were prepared on (100)-oriented STO substrates (nominal 0.1° miscut) by chemical etching in $H_2O:NH_4F:HF$ solution (100:3:1 concentration) and thermal annealing at 950 °C for 2 h. For the bottom electrode, 60 u.c. of $(La_{0.7}Sr_{0.3})MnO_3$ (LSMO) was deposited at 600 °C using $0.9\,mJ\,cm^{-2}$ laser fluence at 0.2 Hz repetition rate in 0.15 mbar $O_2$ atomsphere. PTO ferroelectric barriers were then deposited at 600 °C, 0.2 mbar $O_2$ pressure, $0.45\,mJ\,cm^{-2}$ laser fluence and 4 Hz repetition rate. The layer-by-layer growth of LSMO and PTO films was *in situ* monitored by RHEED, Supplementary Fig. 4a shows the RHEED oscillations measured during the growth. By counting the fringes from the start position, the growth can be stopped after a precise number of layers have been deposited. Supplementary Fig. 4b shows the topography of the surface after the LSMO and PTO layers have been deposited. The top contact Co was then deposited by RF sputtering at $2.5 \times 10^{-3}$ mbar Ar pressure and 20 W applied power. Finally, $40 \times 40\,\mu m^2$ electrodes were patterned into the Co using photolithography and wet-etching of the Co film.

**Electric and magnetic measurements.** $40 \times 40\,\mu m^2$ Co/PTO/LSMO tunnel junction devices have been electrically characterized by measuring current–voltage (I–V) characteristics using Keithley 2635 source-measuring units and an HTTP4 Lake-Shore cryogenic probing station. To switch the polarization orientation of the

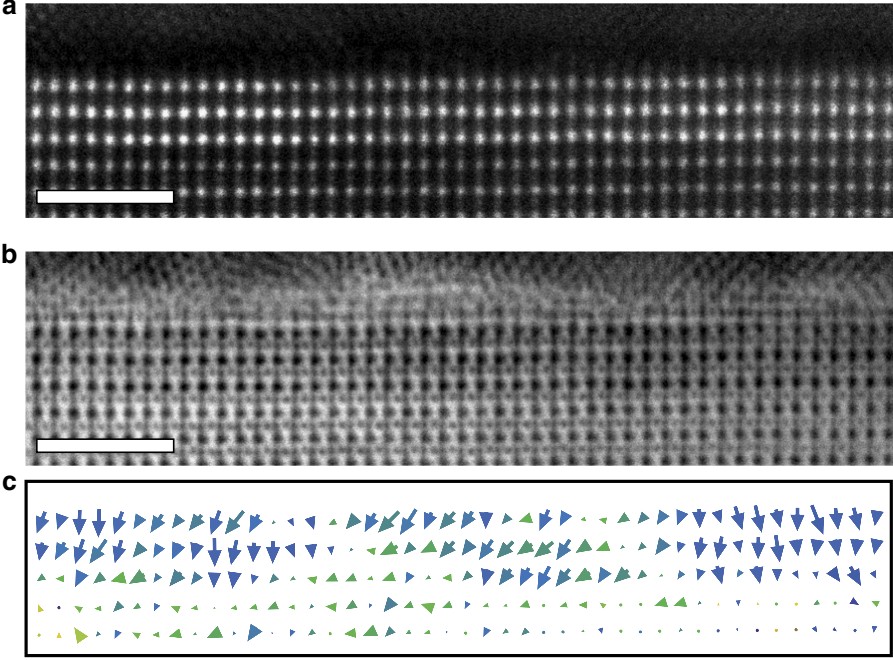

**Figure 4 | Three unit cells thick PTO tunnel junction.** (**a**,**b**) Atomic resolution ADF and ABF images respectively. All Scale bars, 2 nm. (**c**) Corresponding quiver plot of the local dipoles.

barrier, 0.5 ms wide positive/negative voltage pulses of 3 V were applied by using a Tektronix AFG 3102 function generator.

Typical $I$–$V$ characteristics are shown in Supplementary Fig. 5 showing the TER effect. For the 3-u.c. PTO layer, no change in current is observed, showing that the polarization cannot be switched. TMR was evaluated from measuring the tunnelling current with a 200 mV applied bias. Supplementary Fig. 3 shows a change in resistance caused by the magnetization switching in the Co and LSMO electrodes. Macroscopic magnetic hystereses measured on similar, unpatterned samples show that the coercive fields of the patterned devices are very similar to the unpatterned electrodes.

The total magnetic moment of the heterostructure (with the total area $5 \times 5$ mm$^2$) was measured using an Oxford Instruments MagLab vibrating sample magnetometer in the as grown state. All measurements were performed after cooling the samples to 10 K in a $-0.8$ T magnetic field. TMR was calculated as $\frac{R_{\uparrow\uparrow} - R_{\uparrow\downarrow}}{R_{\uparrow\uparrow}}$ where $R_{\downarrow\uparrow}$ and $R_{\uparrow\uparrow}$ are the resistances of the devices with antiparallel and parallel magnetizations of the electrodes.

**Scanning transmission electron microscopy.** All images were taken using a double CEOS corrected (to third order), Schottky emission JEOL ARM-200F microscope operating at 200 kV in STEM mode. ABF images were taken using collection semiangles of 11.5–24 mrad (ref. 28) and the simultaneous ADF signal was collected in the range $\sim 70$–280 mrad. A probe forming convergence semiangle of 21 mrad was used throughout. TEM specimens were prepared using a focused ion beam (FIB) with standard liftout procedures. To reduce contamination, before being inserted into the TEM the specimens were warmed to 50 °C for $\sim 10$ h in a vacuum of $10^{-4}$ Torr. Images in Fig. 2 were acquired as a single scan of $2,048 \times 1,024$ pixels with a dwell time of 20 μs per pixel. For further images, stacks of rapid, successive scans were acquired using the 'StackBuilder' plug-in for Digital Micrograph by Bernard Schaffer (http://dmscript.tavernmaker.de). These were then aligned and averaged using a rigid translation determined from cross correlation with sub-pixel refinement from fitting 3-point parabola. Figure 3d consists of five separate images stitched together before the analysis was performed. Images were taken when a drift below $\sim 5$ pm s$^{-1}$ was measured.

**Polarization analysis.** The local dipoles can be obtained from a displacement of the B-site cation from the octahedra centre, $\Delta_{\text{B}}$, is proportional to the spontaneous polarization, $\mathbf{P}_{\text{S}}$ (ref. 29) This gives

$$\mathbf{P}_{\text{S}} = \kappa \Delta_{\text{B}} \qquad (1)$$

where $\kappa$ is a constant of proportionality that may be deduced from bulk measurements[30]. $\Delta_{\text{B}}$ was measured from ABF images by first filtered with a median filter (3 pixel window) and a Gaussian convolution ($\sigma = 1$ pixel) so that a local pixel maxima (compared with the surrounding 8 pixels) could be found. Positions were then refined using non-linear least squares fitting of a two-dimensional Gaussian function and a modified peak pairs[31] algorithm was used to split the B-site and oxygen sublattices. For each B-site, the centre of mass of the two nearest oxygen atom columns was used to give the centre of the oxygen octahedra. Quantitative polarization values were calculated as

$$\mathbf{P}_{\text{S}} = \frac{1}{\upsilon}(3\boldsymbol{\delta}_{\text{O}}\bar{Z}_{\text{O}} - \boldsymbol{\delta}_{\text{B}}Z_{\text{B}}) \qquad (2)$$

where $\upsilon$ is the volume of the unit cell, $\boldsymbol{\delta}_{\text{O, B}}$ are the displacements of the O and B-site atoms from the positions of centrosymmetry (here taken as the centre of the nearest A-site positions) and $Z_{\text{O, B}}$ are the effective charges of the O and B atoms (taken from ref. 32).

Supplementary Fig. 6a demonstrates the measured displacement schematically where the displacement vector is from the centre of mass of the oxygen atoms to the B-site position. Supplementary Fig. 6b shows this displacement experimentally determined where the displacement is already quite visible.

Previous studies investigating the measurement of atom column positions from ADF–STEM have claimed precisions of $<5$ pm (refs 23,33). By aligning rapidly acquired scans, the precision of the measurements presented here should be within the same range. The measurements can also be greatly affected by small sample tilts that will always be present[34]. To gauge the effects of sample tilt, the LSMO layers were used as a reference where the average oxygen displacement is expected to be zero. No average displacement is observed in the LSMO near to the PTO interface in Figs 2–4, therefore the tilt effects in images are believed to be minimal.

**Data availability.** The data that support the findings of this study are available from the corresponding author upon request.

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

## Acknowledgements

J.J.P.P. and G.A. are supported by the EPSRC (EP/L505110/1). M.A. acknowledges the Wolfson Research Merit Award of the Royal Society. The support of Dietrich Hesse is greatly appreciated.

## Author contributions

J.J.P.P. performed the STEM preparation, experiments and analysis under supervision of A.M.S. and R.B.; G.A. performed the growth and electrical measurements with supervision from M.A.; J.J.P.P. and M.A. wrote the paper with input from all authors.

## Additional information

**Competing financial interests:** The authors declare no competing financial interests.

