## [Peer Review File · Nature Communications]

Reviewers' comments:

Reviewer #1 (Remarks to the Author):

In this work, the authors report, by using aberration-corrected STEM imaging, thickness-dependent domains and domain wall evolutions in PbTiO₃ films sandwiched through asymmetrical electrodes. This work may be of interest for the ferroics community.

However, the present domains and domain wall evolutions do not support the title and the topic (flux closures) being discussed in this work. First, the method used here is flawed. Using O²⁻-Pb²⁺ displacement here to identify local dipole is not reasonable. From <110> projections, local dipoles in PbTiO₃ could be better estimated by Ti⁴⁺ ion displacement with respect to the centre of its two nearest O²⁻ neighbors, since the valence state of Ti⁴⁺ is obvious larger than Pb²⁺, even by considering the effective charge valence (Science 331, 1420 (2011)). For PbTiO₃ lattice without complex polarization curling, using O²⁻-Pb²⁺ displacement only is enough (Nature Mater. 7, 57 (2008)). For so-called flux closures reported here, using O²⁻-Pb²⁺ displacement only is not enough for identifying local dipoles; instead O²⁻-Ti⁴⁺ displacement is needed. If the O²⁻-Pb²⁺ and O²⁻-Ti⁴⁺ displacement give similar dipole vector maps, then it is more convincing that flux closures occur. Even though, the evidence of so-called flux closure reported here is too weak.

Second, it should be careful when using 'toroidal ferroelectricity' since this phrase refers to special 3D topological ferroelectric bubble order, where the vortex cores are aligned along a closed path. Obviously, this type of ferroelectric ordering cannot be directly observed according to a single STEM image.

Third, the authors claimed that '.....in films without electrodes or sandwiched between insulating layers. As such configurations are not suitable for electronic devices, it is wise to investigate if the curling polarisation structures are innate to ferroelectricity or only induced by the absence of electrodes.' This is overstated since the flux closure or vortex sandwiched between insulating SrTiO₃ can be easily integrated into electric devices, where only a bottom electrode is enough (Nature 530, 198 (2016)).

Additionally, I note the imaging condition used by the authors here is described as 'ADF images which were taken using collection semiangles of 11.5-24 mrad and the simultaneous ABF signal was collected in the range ~70-280 mrad.'. It seems that the authors were confused about these imaging conditions, since it is more likely that the ADF is ~70-280 mrad, and ABF is 11.5-24 mrad. Furthermore, the authors claimed that 'Contrast corresponding to the atom columns in the vortices did not show any elongation so there is no evidence of mixed displacements in projection, therefore no other component of dipole rotation is present (as would occur in Néel or Bloch-type walls)'. In an (S)TEM image, it is always difficult to judge atom displacement along the projection directions, which is not necessarily linked with atom column elongation. So in any case, one should be careful when discussing atom displacements along the projection directions.

Reviewer #2 (Remarks to the Author):

This manuscript nicely summarizes information from high resolution transmission electron microscopy images of ultrathin ferroelectric tunnel junctions, in which the relative shifts of cation and anion lattices have been measured directly. Ultimately, the oxygen displacement maps suggest rather complex dipole patterns; while vortex and closure states are expected in systems without electrodes, such complex dipole patterns in structures with electrodes have been less well explored. Given the growth of FTJ research and the potential for FTJs in novel devices, knowledge of the actual polar state associated with the giant electroresistive effect and magnetically tuned tunneling effects could be really important. This study is therefore filling an important gap in the literature.

In principle, I think the manuscript could be published in Nat Comms, but there are important issues that need to be addressed by the authors:

(i) the functional properties of the tunnel junctions are presented and it is clear that the TMR behaviour, for example, is strongly dependent on the polar state expected in the film. However, it is also clear from the direct imaging that the polar state which exists is not the monodomain state imagined in cartoon cross-sections with either "up" or "down" polarisation. I think the manuscript would be much more valuable if the dipole patterns for nominally up and down states in the tunnel junctions could be compared.....what is the actual difference in local polarisation pattern responsible for the differences in tunnel currents ? This would be a very significant element for the work.

(ii) a lot of the rationalisation for the dipole pattern behaviour relies on screening behaviour in the electrodes. A much more serious and quantitative discussion of the relative differences in screening lengths associated with the electrodes is needed.

(iii) there are unsubstantiated conjectures on flexoelectric and Bloch line physics which I do not think are supported by the data - conjecture and hyperbolic discussion should be avoided.

(iv) from what I can tell, the quiver plots show the net oxygen displacements - given that both the cation and anion sublattices have been imaged, it would be MUCH more useful if the authors could plot the local dipole instead (given effective charge assumptions etc).

Minor points: Junquera and Ghosez (Nature 2003) should probably be included as it concerns the role of depolarising fields even when electrodes are present; the title is a little misleading - I'm not even sure what toroidal ferroelectricity is and, even if it does exist, I'm not sure it has been evidenced in this manuscript.

REVIEWERS' COMMENTS:

Reviewer #1 (Remarks to the Author):

I find the authors well addressed (point-by-point)the concerns raised by the reviewer. I am happy to recommend the revised version for publication in Nature Communications, although I still think it would be nice if the authors could carry out phase-field modeling to extract more physical insight on the structural evolution via film thickness.

In the caption of figure 3(d), one "ABF" should read "ADF".

Reviewer #2 (Remarks to the Author):

The authors have considered the reviews maturely and sensibly and made appropriate changes to the text and the analysis. I'm content that this can now be published.

We thank both reviewers for the careful reading and extremely useful comments and suggestions. We have addressed all of them, as detailed below, and we believe our paper has gained in clarity and quality.

Reviewer 1

In this work, the authors report, by using aberration-corrected STEM imaging, thickness-dependent domains and domain wall evolutions in PbTiO₃ films sandwiched through asymmetrical electrodes. This work may be of interest for the ferroics community.

However, the present domains and domain wall evolutions do not support the title and the topic (flux closures) being discussed in this work.

1. First, the method used here is flawed. Using O₂--Pb²⁺ displacement here to identify local dipole is not reasonable. From <110> projections, local dipoles in PbTiO₃ could be better estimated by Ti⁴⁺ ion displacement with respect to the centre of its two nearest O₂- neighbours, since the valence state of Ti⁴⁺ is obvious larger than Pb²⁺, even by considering the effective charge valence (Science 331, 1420 (2011)). For PbTiO₃ lattice without complex polarization curling, using O₂--Pb²⁺ displacement only is enough (Nature Mater. 7, 57 (2008)). For so-called flux closures reported here, using O₂--Pb²⁺ displacement only is not enough for identifying local dipoles; instead O₂--Ti⁴⁺ displacement is needed. If the O₂--Pb²⁺ and O₂--Ti⁴⁺ displacement give similar dipole vector maps, then it is more convincing that flux closures occur. Even though, the evidence of so-called flux closure reported here is too weak.

We have modified Figs. 2-4 in response to both this comment and that of the second referee. (In fact, we had already looked at this point before submitting the paper, and found little difference between the two results). For a more general audience interested in functional properties, we agree that it is better to display polarisation to avoid any doubt. The vortex structure remains quite clear in Fig.2 and is clear evidence of flux closure. We also note that the measurement of polarisation that we obtain is very similar to the calculated bulk value ($\sim 84 \mu\text{Cm}^{-2}$), giving confidence in the measurement that is absent in some other papers (e.g. Fig. 6 in 10.1038/nmat1808)

2. Second, it should be careful when using 'toroidal ferroelectricity' since this phrase refers to special 3D topological ferroelectric bubble order, where the vortex cores are aligned along a closed path. Obviously, this type of ferroelectric ordering cannot be directly observed according to a single STEM image.

We agree that it is not possible to unequivocally establish the 3D nature of the flux vortices from our STEM data and have therefore removed this phrase from the title. Nevertheless, one would expect these vortices to have a continuity in three dimensions and their association with domain walls strongly suggests that they should form closed loops (since a domain must have a closed boundary).

3. Third, the authors claimed that '.....in films without electrodes or sandwiched between insulating layers. As such configurations are not suitable for electronic devices, it is wise to investigate if the curling polarisation structures are innate to ferroelectricity or only induced by the absence of electrodes.' This is overstated since the flux closure or vortex sandwiched between insulating SrTiO₃ can be easily integrated into electric devices, where only a bottom electrode is enough (Nature 530, 198 (2016)).

Indeed a PTO/STO superlattices structure is grown in the cited paper on a bottom SRO electrode. The whole structure is about 100 nm thick and each individual PTO layer is sandwiched between two STO layers and not a metal. This maximises, in our opinion, the effect of depolarising field. It is

indeed possible to have a top electrode onto the entire superlattice structure, but the individual layers will still be under high depolarising field since there will be no charge at the STO/PTO boundary to compensate the polarisation.

Since we agree that such structures can be integrated in electronic devices, we modify the sentence as follows: "As such configurations mitigate the role of screening charges provided by metal electrodes in metal-ferroelectric-metal structures commonly used as electronic devices, it is wise to investigate if the curling polarisation structures are innate to ferroelectricity, or only induced by the absence of electrodes.

4. Additionally, I note the imaging condition used by the authors here is described as 'ADF images which were taken using collection semiangles of 11.5-24 mrad and the simultaneous ABF signal was collected in the range ~70-280 mrad.'. It seems that the authors were confused about these imaging conditions, since it is more likely that the ADF is ~70-280 mrad, and ABF is 11.5-24 mrad.

This was simply a typographical error and has been corrected. We thank the reviewer for spotting it.

5. Furthermore, the authors claimed that 'Contrast corresponding to the atom columns in the vortices did not show any elongation so there is no evidence of mixed displacements in projection, therefore no other component of dipole rotation is present (as would occur in Néel or Bloch-type walls)'. In an (S)TEM image, it is always difficult to judge atom displacement along the projection directions, which is not necessarily linked with atom column elongation. So in any case, one should be careful when discussing atom displacements along the projection directions.

This is a misunderstanding (which indicates a problem with our text, we have corrected this as below). We do not make any claims about measuring displacements along the beam direction (and agree this cannot be done from a STEM image). Our point is that displacements in an atom column, perpendicular to the beam direction, that vary through the specimen will produce an elongated appearance (see e.g. Jia et al. Science 331 (2011), p1420). We have changed the text to 'We note that an atom column that contains varying displacements perpendicular to the beam direction would appear elongated (i.e. corresponding to the projection of the mixed positions). Such an effect is absent, indicating that the measured dipoles are constant through the thickness of the specimen. *This would not be the case for Néel or Bloch-type walls extending within the thickness of the TEM specimen.*'

Reviewer 2

This manuscript nicely summarizes information from high resolution transmission electron microscopy images of ultrathin ferroelectric tunnel junctions, in which the relative shifts of cation and anion lattices have been measured directly. Ultimately, the oxygen displacement maps suggest rather complex dipole patterns; while vortex and closure states are expected in systems without electrodes, such complex dipole patterns in structures with electrodes have been less well explored. Given the growth of FTJ research and the potential for FTJs in novel devices, knowledge of the actual polar state associated with the giant electroresistive effect and magnetically tuned tunneling effects could be really important. This study is therefore filling an important gap in the literature.

In principle, I think the manuscript could be published in Nat Comms, but there are important issues that need to be addressed by the authors:

We thank the reviewer for the positive review. We are addressing the issues raised below.

1. the functional properties of the tunnel junctions are presented and it is clear that the TMR behaviour, for example, is strongly dependent on the polar state expected in the film. However, it is also clear from the direct imaging that the polar state which exists is not the monodomain state imagined in cartoon cross-

sections with either "up" or "down" polarisation. I think the manuscript would be much more valuable if the dipole patterns for nominally up and down states in the tunnel junctions could be compared.....what is the actual difference in local polarisation pattern responsible for the differences in tunnel currents ? This would be a very significant element for the work.

We agree that the global properties of the tunnel junctions will of course depend upon the polarisation state in the PTO. As for the TMR that is specifically addressed by this question the behaviour is not much changed by the ferroelectric switching, as shown in Fig. S3 of the manuscript. The results are quite different from the previous case (ref. 12 of MS) when the ferroelectric barrier was PZT and switching of ferroelectric polarisation massively changed the TMR. Obviously further studies, most probably synchrotron-based, are needed to fully understand this behaviour. Nevertheless, the material in this paper was not poled, and would therefore be expected to have a mixed domain structure, as we observe. The extent to which complete poling is possible is a very important question, but is not addressed in this paper (although we are working on this at present). We have added the word 'unpoled' both in the abstract and throughout the text to clarify this point.

2. a lot of the rationalisation for the dipole pattern behaviour relies on screening behaviour in the electrodes. A much more serious and quantitative discussion of the relative differences in screening lengths associated with the electrodes is needed.

The screening mechanism of the polarisation is a rather complicated issue to address by a "simple" TEM analysis. We have performed nevertheless an analysis of the polarisation distribution at the PTO/LSMO interface. As can be seen in the new figure (Fig. S6) in the supplementary information the polarisation extends within the LSMO layer over a few unit cells. This is consistent with our previous x-ray-based study on PZT/LSMO interface (Preziosi, Alexe, Hesse, Saluzzo, PRL, 2015) which showed that the ferroelectric polarisation induced a more complex reconstruction of the bonding within the LSMO than a simple carrier depletion/accumulation, as initially assumed. Due to the polycrystalline nature of the top Co electrode, a similar analysis is unfortunately not possible. We can only mention that the polarisation drops to a lower value exactly at the PTO/Co interface than at the PTO/LSMO interface. This points to a different screening mechanism at Co than at LSMO, which in principle is expected. A detailed analysis of the screening mechanism and the two interfaces is rather difficult at this moment and is not within the scope of the present paper.

We have nevertheless introduced the figure below in the supplementary information and a comment related to the topic within the main text: "It can be seen in Fig. S6 that the polarisation is mostly constant across the film but extends into the LSMO layer, inducing a displacement within the first 2-3 u.c. of the interface."

3. there are unsubstantiated conjectures on flexoelectric and Bloch line physics which I do not think are supported by the data - conjecture and hyperbolic discussion should be avoided.

We agree with the reviewer that the discussions related to Bloch lines are rather speculative, thus we removed any reference to it. In the same time, it is clear that as the thickness of the PTO layer decreases under 9 uc. the system crosses a point from a periodic domain pattern, as described by Fong et al. PRL, to a rather disordered state but with a certain remnant periodicity. It is why we defined this phase as an incommensurate phase. A mean-field theoretical description of a similar phase transition in a pure ferroelastic system has been very recently given by Pottker and Salje. We feel that a reference to this work is appropriate in the context of our work. Nevertheless, while the flexoelectricity was the driving force for the incommensurate phase in the Pottker and Salje work, here indeed some other driving forces, such as an inhomogeneous depolarising field given by the polycrystalline Co top layer, might be the real driving force. We have modified the relevant paragraph as:

“The driving force for this is the flexoelectric effect. Indeed our investigated PTO is both ferroelectric and ferroelastic and, under a certain thickness, **additional parameters such as inhomogeneous electric fields** might induce crossing of the Lifshitz point and drive the system into an incommensurate phase. ~~It is worth noting that a strong in-plane misfit strain, as is the case for BaTiO₃ on NdGaO₃, might mitigate the flexoelectric effect and stabilise the simple paraelectric-ferroelectric phase transition down to 2 unit cell film thicknesses.²⁵ In the present case the misfit between PTO and the LSMO is only 0.6%, potentially enhancing the influence of flexoelectric coupling.⁸~~

4. from what I can tell, the quiver plots show the net oxygen displacements - given that both the cation and anion sublattices have been imaged, it would be MUCH more useful if the authors could plot the local dipole instead (given effective charge assumptions etc).

This has been addressed in the response to referee 1.

5. Minor points: Junquera and Ghosez (Nature 2003) should probably be included as it concerns the role of depolarising fields even when electrodes are present; the title is a little misleading - I'm not even sure what toroidal ferroelectricity is and, even if it does exist, I'm not sure it has been evidenced in this manuscript.

First point – We have added the reference in our discussion of the depolarising field in ultrathin films.

Second point – the term ‘Toroidal Ferroelectricity’ has been removed from the title, as mentioned in our response to point 2 of referee 1.